# Satisfying Complex User Needs: M³Agent for Conversational Multi-Item Recommendation

## Abstract

Fulfilling complex, multi-item, multi-constraint user requests in conversational commerce is a critical and largely unsolved research problem. Existing paradigms, from traditional recommenders to modern LLM-based agents, fail due to a crisis of grounding and an inability to handle trade-off complexity. To address these failures, we introduce M³Agent , an agentic framework that bridges natural language and grounded, optimal recommendations. We are the first to reformulate this task as a unified multi-objective optimization problem, where the agent's plan is the provably optimal solution to a holistic objective. M³Agent 's cognitive architecture employs two core mechanisms: an Split-Prune Constraint Tree for grounded, interactive memory, and a Pareto-complete search to find all optimal trade-offs between competing quality objectives. Experiments on two large-scale, real-world datasets show consistent and significant gains over strong baselines, demonstrating that M³Agent effectively translates free-text multi-item requests into catalog-valid, requirement-satisfying recommendations.

## 1 Introduction

Modern recommender systems are a cornerstone of e-commerce Schafer et al. (1999); Covington et al. (2016); Wu et al. (2023); Yang et al. (2025). As platforms integrate conversational AI, the frontier is shifting from simple information retrieval to complex, goal-oriented planning. Users increasingly expect digital assistants to act not as search engines, but as competent planners that can understand nuanced, evolving goals and deliver coherent, actionable solutions. This

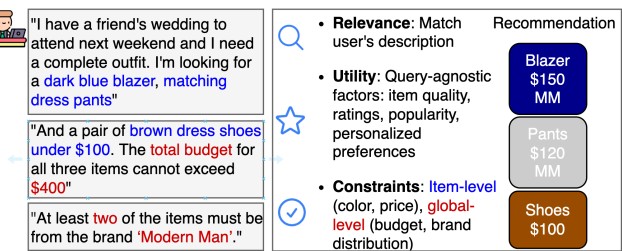

Figure 1: A multi-item recommendation scenario.

translates to a critical and largely unsolved research problem: fulfilling multi-item, multi-constraint user requests within a dynamic, real-world environment.

The complexity of this task becomes apparent in a typical conversational scenario, as shown in Figure 1. A user's intent unfolds across multiple turns, culminating in a request that is not a simple query, but a sophisticated planning specification: "I have a friend's wedding to attend next weekend and I need a complete outfit. I'm looking for a dark blue blazer, matching dress pants. And a pair of brown dress shoes under $100. The total budget for all three items cannot exceed $400". At least two of the items must be from the brand 'Modern Man'."

Successfully fulfilling this request is about formulating a valid plan that satisfies a trilemma of competing objectives: the explicit textual descriptions (**Relevance**), the user's long-term profile preferences (**Utility**), and a complex web of per-item and cross-item rules (**Constraint Satisfaction**). These objectives are in inherent tension—a cheaper item may satisfy the budget but have lower relevance; a perfectly relevant item might conflict with long-term utility. A system that cannot navigate these trade-offs in a principled way is destined to fail.

**The Failure of Existing Paradigms.** Current approaches are ill-equipped for this challenge. Traditional machine learning recommenders learn historical behaviors to predict future interests Bai et al. (2019) by optimizing collaborative-filtering objectives. They are fundamentally designed to echo a user's past, rendering them ineffective at prioritizing a user's immediate, explicit, and often novel requirements expressed in natural language. The advent of Large Language Models (LLMs) and the agent systems integrating them brings powerful language understanding He et al. (2023); Lin et al. (2025), but introduces critical new failures. The first is a dual crisis of **grounding and brittleness**. LLM agents that directly output recommendations face the **grounding crisis**: an LLM's knowledge is static and has no access to the dynamic, real-time state of a **live catalog**, leading to "hallucinated" recommendations of unavailable products. Agent systems that chain together independent modules are fundamentally brittle; an early-stage parsing error often leads to an empty result set, and complex queries can cause a "Lost-in-the-Middle" phenomenon. Critically, these systems lack a robust, structured memory to accumulate and reason about constraints over a multi-turn dialogue. The second is a crisis of **trade-off and control**. An LLM might understand the wedding outfit request, but it lacks a principled mechanism for balancing multiple objectives. This leads to uncontrolled, "all-or-nothing" behavior: systems either rigidly enforce every constraint—often returning nothing—or arbitrarily ignore some requirements, violating the user's conditions. They cannot reason about trade-offs, nor can they make a principled final choice from a set of valid options.

The limitations of existing paradigms are symptoms of a flawed underlying abstraction. The path forward requires a conceptual shift away from the passive "pipeline" and toward a new abstraction: an active, reasoning planning agent. Such an agent must be able to formulate a holistic strategy, negotiate trade-offs, and dynamically adapt its plan based on new information from both the user and the live catalog.

To realize this, we introduce $M^3$AGENT (Multi-item, Multi-constraint, Multi-objective Agent), a conversational framework that acts as an interactive bridge between users and product catalogs. We are the first to reformulate multi-item recommendation as a single, unified **multi-objective optimization** problem. This formulation is not merely a tool the agent uses; it is the agent's cognitive architecture, allowing it to reason about competing goals in a principled way.

The agent's task is executed in two steps. First, it overcomes the grounding and brittleness crisis using an **Split-Prune Constraint Tree**, a novel data structure that serves as its working memory to translate unstructured language into a feasible solution space grounded in the live catalog. Within this space, it resolves the trade-off and control crisis by performing a **Pareto-complete search** to find the entire set of optimal trade-offs between relevance, utility, and constraint satisfaction. Second, having identified all non-dominated solutions, the agent applies a hierarchical selection policy to choose the final recommendation. This entire process is interactive, reframing the multi-turn dialogue as a principled loop of continuous, iterative optimization.

**Summary of Contributions:**

• **Unified Multi-Objective Formulation:** We are the first to reformulate multi-item recommendation with constraints as a multi-objective optimization problem, yielding an agent that plans and optimizes holistically.
• **Agentic Planning and Adaptation via Split-Prune Constraint Tree:** We design $M^3$AGENT as an interactive planner that uses the Split-Prune Constraint Tree as memory to dynamically balance user requirements and catalog realities, enabling robust multi-turn conversational adaptability.
• **Theoretical and Empirical Soundness:** We provide formal theoretical guarantees of efficiency and optimality (Pareto-completeness). We conduct extensive experiments on new, realistic e-commerce benchmarks that we construct and will release, showing that $M^3$AGENT significantly outperforms strong baselines in fulfilling complex, constraint-rich requests.

## 2 RELATED WORK

### 2.1 MULTI-ITEM RECOMMENDATION: FROM BUNDLES TO CONSTRAINED OPTIMIZATION

Recommending multiple items as a coherent set has been explored through slate and bundle recommendation. Early work often pre-defined bundles and matched them to users Hu et al. (2023); Bai et al. (2019), a paradigm limited by the combinatorial explosion of item sets. More flexible approaches

dynamically construct bundles using GNNs or Transformers to model item complementarity based on user history Yang et al. (2021); Li et al. (2023). Recent work has shifted toward generative models, such as diffusion models that can create an entire slate from a textual prompt (Tomasi et al., 2025), and has highlighted the need for models to be intent-aware to generate high-quality bundles under realistic constraints (Sun et al., 2024). However, these approaches are fundamentally designed to optimize for fuzzy, data-driven objectives like "complementarity," not for the hard, logical constraints common in user requests. Realizing that real-world scenarios require balancing multiple competing criteria has led to the rise of multi-objective recommender systems (Jannach & Abdollahpouri, 2023). This includes casting recommendation as a Pareto-optimal trade-off selection problem to navigate between objectives like relevance and diversity (Jeunen et al., 2024), or as a bi-level optimization in conversational settings (Chu et al., 2023). While these constraint-aware frameworks are powerful, they typically assume that the objectives and constraints are well-defined and given to the system, and do not address the crucial preceding step: how to reliably extract, interpret, and ground a complex set of objectives and constraints from ambiguous natural language in a dynamic environment.

## 2.2 Conversational Recommendation and Tool-Augmented LLM Agents

The rise of LLMs has inspired a new wave of agentic systems for conversational recommendation, but naive application often yields suboptimal results (Wang et al., 2023b). To create more capable agents, research has advanced their abilities in several complementary directions. A primary focus has been on improving the internal reasoning and planning processes that structure the agent's behavior. Frameworks like Chain-of-Thought (CoT) Wei et al. (2022) and Tree-of-Thoughts (ToT) Yao et al. (2023a) enhance step-wise reasoning, while multi-agent designs explicitly control dialogue acts to keep the conversation goal-directed (Fang et al., 2024). A crucial evolution of this is interleaving reasoning with actions, as seen in the ReAct paradigm (Yao et al., 2023b), where the agent can decide to call external APIs and incorporate the results. This has led to a rich ecosystem of tool-augmented agents that can ground their reasoning by interacting with external knowledge sources (Schick et al., 2023; Qin et al., 2023), with some even planning tool usage under budget constraints (Zheng et al., 2024). To address the LLM's context limitations in long dialogues, another line of work has focused on equipping agents with structured memory, allowing them to persist and retrieve knowledge across turns (Park et al., 2023; Wang et al., 2023a). While these are crucial steps toward creating more capable systems, a fundamental gap remains. The enhancement of abstract reasoning often leaves the agent's plans disconnected from the real-world, ground-truth state of a live product catalog. Similarly, augmenting agents with tools and memory treats the catalog as a black-box tool to be queried, not as a structured environment to be planned over. These agents can ask "what are the cheapest waterproof shoes?" but lack a formal mechanism to solve the holistic, constrained optimization problem: "what is the best combination of shoes and a jacket that satisfies a global budget and brand quota?"

## 3 Problem Formulation

**Definition 3.1** (The Product Catalog). The product catalog, $\mathcal{P}$, is a finite set of items. Each item $i \in \mathcal{P}$ is a semi-structured record with a set of attribute-value pairs, e.g., $i = \{$id: 123, brand: "Nike", price: 75, ...$\}$. The catalog is considered **live**, meaning its state—such as price, inventory levels, or even the set of available attributes—can change dynamically. New items may be added, and existing ones removed, making the catalog a real-time environment that is not part of any LLM's static, pre-trained knowledge.

Our task is to return multiple items over the **live, structured catalog**, given a natural language query $q$, which specifies a user's intent for $n$ desired items. For example, the query in Fig1 specifies an outfit with $n = 3$ items. The output is formulated as a tuple $y = (y_1, \ldots, y_n)$ where each $y_j \in \mathcal{P}$. The quality of any given tuple is determined by a multi-faceted objective: how relevant the items are to the query, their intrinsic utility, and the degree to which they jointly satisfy all user-specified constraints. The problem is fraught with several fundamental challenges: 1) **Grounding and Feasibility:** Translating natural language into structured constraints is non-trivial. For a query like "dark blue blazer, matching dress pants," the attribute `color` may not have a "dark blue" value in the catalog, and the concept of "matching" is ambiguous (e.g., does it mean the same color or a complementary color?). More critically, the conjunction of all extracted constraints may be too restrictive, resulting in an empty feasible set and thus no recommendation—a critical failure for a

user-facing system. 2) **Computational Complexity for Trade-off and Optimality:** Evaluating all possible tuples to find the optimal one and satisfying as much constraints requires a search space $\mathcal{P}^n$, which is computationally intractable for any non-trivial catalog size $N$ and number of items $n$ and lacks a theoretically sound guarantee

# 4 METHODOLOGY

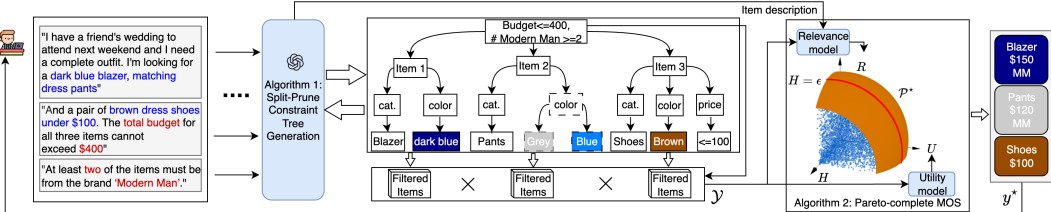

Figure 2: Illustration of the proposed $M^3$AGENT framework with its key components.

The overall architecture of our $M^3$AGENT framework is illustrated in Fig. 2. The framework operates in two stages. First, to address the dual challenges of **grounding** natural language and managing **computational complexity**, we introduce the Split-Prune Constraint Tree (Section 4.2). This agentic algorithm (Algorithm 1) intelligently filters the vast product catalog to construct a small yet robustly feasible plan space, $\mathcal{Y}$. Second, within this constrained space, the agent performs an efficient, Pareto-complete search (Algorithm 2) to find the full Pareto-optimal set $\mathcal{P}^\star$ with respect to the objectives $(R, U, H)$. We provide theoretical guarantees for both stages, ensuring that our overall pipeline is both computationally efficient and Pareto-complete.

## 4.1 $M^3$AGENT : AGENTIC PLANNING AS MULTI-OBJECTIVE OPTIMIZATION

**Pareto-Optimality in Multi-Objective Recommendations**

**Definition 4.1** (Pareto-Optimal Recommendation)**.** Given several objective functions to be maximized, $(f_1(y), \ldots, f_m(y))$, a candidate recommendation $y$ from a feasible set $\mathcal{Y}$ is **Pareto-optimal** if there is no other feasible recommendation $y' \in \mathcal{Y}$ such that $f_i(y') \geq f_i(y)$ for all objectives $f_i$, and strictly greater for at least one objective. In other words, $y$ cannot be *dominated* by any other solution.

**Definition 4.2** (Recommendation Pareto Front)**.** The **Recommendation Pareto front** is the set of all Pareto-optimal recommendations. In our framework, this set is formally defined with respect to our three core objectives:

$$\mathcal{P}^\star = \{\, y \in \mathcal{Y} \,:\, y \text{ is Pareto-optimal w.r.t. the objectives } (R, U, H) \,\}.$$

Each point on the Pareto front represents a different optimal trade-off among the objectives.

To systematically address the trade-offs among the core objectives, our goal is to find solutions that are Pareto-optimal with respect to a vector of three objective functions:

$$\max_{y \in \mathcal{P}^n} \; \Big( R(q, y), U(y), H(y) \Big) \tag{1}$$

where the agent's task is to first find the set of non-dominated solutions $\mathcal{P}^\star$ and then apply a selection policy to choose the final recommendation $y^\star \in \mathcal{P}^\star$. The three components are defined as follows:

**Relevance** ($R$)    The *relevance* term, defined as $R(q, y) = \frac{1}{n} \sum_{j=1}^n \text{rel}(q, y_j, j)$, measures the query-conditioned relevance of the items in the plan. Each $\text{rel}(q, y_j, j)$ quantifies how well item $y_j$ matches the portion of query $q$ pertaining to the $j$-th item. We implement this with a text-matching score, where $\text{rel}(q, i, j) = \text{sim}(\text{LLM}(q, j), \text{desc}(i))$. Here, $\text{LLM}(q, j)$ is an LLM-generated description of the ideal item $j$, and $\text{desc}(i)$ is the catalog text for item $i$.

**Utility** ($U$)   The *utility* term, defined as $U(y) = \frac{1}{n} \sum_{j=1}^{n} \mathrm{util}(y_j)$, measures the query-agnostic utility of the items in the plan. This component encourages the agent to select items that are intrinsically desirable (e.g., popular, highly-rated, or personalized to the user's general tastes) beyond simply matching the query text.

**Constraint Satisfaction** ($H$)   The *constraint satisfaction* term, defined as $H(y) = \sum_{c \in C} w_c \cdot \mathbf{1}_{\{y \text{ satisfies } c\}}$, quantifies how well the plan adheres to explicit user preferences. The set $C$ contains all constraints parsed from the query $q$. This includes **item-level** constraints (e.g., "price: under $50" for a shirt) and **global** constraints that span multiple items (e.g., a total budget). Satisfying a constraint $c$ contributes its confidence weight $w_c$ to the objective, driving the agent to construct a plan that satisfies as many high-confidence preferences as possible. The formal definition of the global constraint set, $\mathcal{G}(q)$, and further examples are detailed in Appendix A.

### 4.2 Stage 1: Split-Prune Constraint Tree for Grounded, Robust and Interactive Planning

The first stage of our pipeline is designed to solve two core challenges: 1) translating ambiguous natural language into a feasible, structured plan grounded in the reality of a live catalog, and 2) reducing the search space to manage computational complexity.

The agent's planning process begins by interpreting the user's request, prompting an LLM with the catalog schema to extract constraints into a hierarchical tree of (`field, value`) pairs (prompts detailed in Appendix D). A crucial grounding step immediately follows: the agent validates every extracted constraint against the live catalog. If a parsed `value` does not exist for a given `field`, or if the LLM's confidence is low, the system intelligently splits the constraint into multiple, similar, and catalog-valid options (`SplitValues` in Algorithm 1). This proactive validation ensures the agent's plan is always rooted in reality.

To guarantee a feasible solution, the agent employs an iterative pruning strategy. If the conjunction of all constraints results in an empty set, the agent systematically relaxes the plan by removing the constraint with the lowest confidence score. This confidence, $w_c$ is calculated as the geometric mean of token probabilities for the entire (`field, value`) segment, allowing the agent to make a principled decision to drop the most uncertain part of the user's request. This loop continues until a non-empty feasible plan is found, transforming a potential failure into a robust recommendation.

Finally, the constraint tree serves as the agent's dynamic conversational memory. When a user expresses new requirements, the agent perceives this as an update to its goals, modifying the existing tree and re-initiating the planning loop to find a new optimal solution.

---

**Algorithm 1:** Split-Prune Constraint Tree

---

**Input:** query $q$; token-level probabilities $P$; catalog filter $\Gamma$; threshold $\theta$; max iterations $k_{\max}$
**Output:** constraint tree $\mathcal{T}^*$

---

1 $\mathcal{T} \leftarrow$ InitTree();   $\mathcal{C} \leftarrow$ ExtractConstraints($q$);
2 **foreach** $c = (field, value) \in \mathcal{C}$ **do**
3      $s_{\text{field}} \leftarrow$ ScoreToken($field$; $P$);   $s_{\text{value}} \leftarrow$ ScoreToken($value$; $P$);
4      **if** $s_{field} \geq \theta$ **then**
5          InsertField($\mathcal{T}$, $field$);
6          **if** $s_{value} \geq \theta$ **then**
7              InsertValue($\mathcal{T}$, $field$, $value$)
8          **else**
9              SplitValues($\mathcal{T}$, $field$, Candidates($value$))

10 $k \leftarrow 0$;   $\mathcal{R} \leftarrow \Gamma(\mathcal{T})$;
11 **while** $\mathcal{R} = \emptyset$ **and** $k < k_{\max}$ **do**
12      $u \leftarrow$ LowestConfidenceNode($\mathcal{T}$);   Prune($\mathcal{T}$, $u$);
13      $k \leftarrow k + 1$;   $\mathcal{R} \leftarrow \Gamma(\mathcal{T})$;
14 **return** $\mathcal{T}^* \leftarrow \mathcal{T}$

---

The final tree $\mathcal{T}^*$ encodes a set of filtered candidate pools $C_j(q)$ for each item position $j$ and a set of global constraints $\mathcal{G}(q)$. From this, we construct the finite set of all feasible tuples, $\mathcal{Y}$, as follows:

$$\mathcal{Y} \leftarrow \{y = (y_1, \ldots, y_n) \mid \forall j, y_j \in C_j(q) \text{ and } y \text{ satisfies } \mathcal{G}(q)\}$$

This set $\mathcal{Y}$ then serves as the input to the second stage of our pipeline, the multi-objective search.

**Theorem 4.3** (Computational Efficiency). *Let the size of the feasible set derived from the constraint tree be bounded by $M = |\mathcal{Y}|$. The overall time complexity of our pipeline to compute the Pareto-optimal set $\mathcal{P}^\star$ is $O(N + M)$, where $N$ is the total number of items in the catalog. This represents an exponential speedup over exhaustive search methods, which have a complexity of $\Omega(N^n)$.*

*Proof Sketch.* The overall $O(N + M)$ complexity arises from two terms. The $O(N)$ term stems from the initial linear scan of the catalog to populate the candidate pools $\{C_j(q)\}$. The $O(M)$ term bounds the subsequent search, where at most $M = \prod_j |C_j(q)|$ combinations are evaluated against the global constraints $\mathcal{G}(q)$ to form the feasible set $\mathcal{Y}$ and find the Pareto front. This separation of filtering and search provides an exponential speedup over the $\Omega(N^n)$ complexity of exhaustive methods. The full proof is deferred to Appendix B. $\qquad\square$

### 4.3 STAGE 2: PARETO-COMPLETE MULTI-OBJECTIVE SEARCH

Once the feasible plan space $\mathcal{Y}$ is defined by the constraint tree, the second stage of our pipeline searches within this space to find the full set of Pareto-optimal solutions with respect to the objectives $(R, U, H)$. This is accomplished by the procedure detailed in Algorithm 2. The Pareto-completeness is formally established in the proof of Theorem 4.4.

---

**Algorithm 2:** Pareto-Complete Multi-Objective Search

**Input:** Feasible set of tuples $\mathcal{Y}$
**Output:** Pareto-optimal set $\mathcal{P}^\star$; Recommended tuple $y^*$

1  $\mathcal{P}^\star \leftarrow \emptyset$ Let $\mathcal{E}_H$ be the set of all distinct $H$-values in $\mathcal{Y}$;
2  **foreach** $\epsilon \in \mathcal{E}_H$ **do**
3  $\quad$ Let $S_\epsilon \leftarrow \{y \in \mathcal{Y} : H(y) \geq \epsilon\}$;
4  $\quad$ $\mathcal{P}_\epsilon \leftarrow$ COMPUTEPARETOFRONT$(\{(R(y), U(y)) \mid y \in S_\epsilon\})$;
5  $\quad$ **foreach** $y \in \mathcal{P}_\epsilon$ **do**
6  $\quad\quad$ **if** $\nexists z \in \mathcal{P}^\star$ *s.t. $z$ dominates $y$* **then**
7  $\quad\quad\quad$ $\mathcal{P}^\star \leftarrow \{z \in \mathcal{P}^\star \mid z \text{ is not dominated by } y\} \cup \{y\}$;

8  $y^* \leftarrow \arg\max_{y \in \mathcal{P}^\star} \big(H(y), R(q, y) + \lambda_U U(y)\big)$ **return** $y^*, \mathcal{P}^\star$

---

**Theorem 4.4** (Pareto-Completeness of Search). *Let $\mathcal{Y}$ be the finite feasible set of tuples derived from Algorithm 1. The set $\mathcal{P}^\star$ returned by Algorithm 2 contains every Pareto-optimal tuple $y^\dagger \in \mathcal{Y}$ with respect to the objectives $(R, U, H)$.*

*Proof Sketch.* The proof's key insight is that the 3D Pareto-optimality problem can be decomposed into a series of independent 2D subproblems, one for each level of the objective $H$. Our algorithm's hierarchical search structure mirrors this decomposition perfectly. By finding the complete Pareto front for each 2D subproblem, overall completeness is guaranteed. The full proof is deferred to Appendix C. $\qquad\square$

The final recommendation produced by our system is therefore lexicographically optimal with respect to our defined objectives, a guarantee that follows from the Pareto-completeness of our search stage. This unified framework finds high-quality recommendations that are not only Pareto-optimal within a robustly defined feasible set, but are also selected based on a clear, hierarchical preference for user-specified constraints.

# 5 EXPERIMENT

## 5.1 DATASETS AND EXPERIMENTAL SETUP

**Datasets**   We evaluate on two datasets: H&M (Borgersen et al., 2024) (20,000 users, 105,542 fashion items) and Amazon (Jin et al., 2023) (10,000 users, 307,721 multi-category items). Both datasets feature users with shopping history sequences. For each user, we randomly select 1-5 items from their history and use Claude-Sonnet-4 to generate user queries (prompts detailed in Appendix D).

**Evaluation metrics.**   We evaluate our models using multiple metrics at cutoff points (1, 3, and 5): Hit Rate@k measures whether at least one relevant item appears in the top-k recommendations; NDCG@k evaluates both relevance and ranking quality by discounting relevant items lower in the list; Recall_macro@k (average per-user) and Recall_micro@k (global) assess the fraction of relevant items retrieved; Perfect Hit Rate@k checks if all relevant items are included in the top-k. Details are provided in Appendix E.

**Baseline methods**   To evaluate the effectiveness of our $M^3$AGENT approach, we compare against several baseline methods: **History**: Randomly samples items from the user's historical purchase history. **KNN**: Computes text embedding similarities between user queries and item descriptions to rank candidates by semantic relevance. **ChatREC** Gao et al. (2023): Uses large language models to combine user query understanding with profile-based preferences. **Ideal utility model**: Assumes perfect knowledge of query-agnostic item quality by assigning maximum scores to items the user will purchase. **InteRecAgent** Huang et al. (2025): Combines LLM-driven tool usage, including database queries, similarity searches, and utility ranking pipelines for multi-step recommendations.

To ensure fair comparison, all similarity-based methods use the multilingual-e5-large model for embeddings and FAISS for retrieval, all LLM-based methods employ Qwen2.5-7B-Instruct.

**Utility model impact**   Real-world recommender systems employ a variety of utility models, from tree-based models to transformers. To isolate the impact of the utility model, we evaluate each method both with and without the Ideal utility model. Any real-world utility model's performance should in the middle of the two.

## 5.2 PERFORMANCE COMPARISON

Table 1: Performance comparison on H&M and Amazon datasets

| Method | Hit Rate@k | | | NDCG@k | | | Recall_macro@k | | | Recall_micro@k | | | Perfect Hit Rate@k | | |
|---|---|---|---|---|---|---|---|---|---|---|---|---|---|---|---|
| | @1 | @3 | @5 | @1 | @3 | @5 | @1 | @3 | @5 | @1 | @3 | @5 | @1 | @3 | @5 |
| **H&M Dataset** | | | | | | | | | | | | | | | |
| *Methods without ideal utility model* | | | | | | | | | | | | | | | |
| History | 0.0087 | 0.0166 | 0.0203 | 0.0022 | 0.0033 | 0.0036 | 0.0036 | 0.0068 | 0.0083 | 0.0034 | 0.0066 | 0.0081 | 0.0011 | 0.0019 | 0.0025 |
| KNN | 0.0360 | 0.0776 | 0.1109 | 0.0193 | 0.0313 | 0.0380 | 0.0230 | 0.0491 | 0.0698 | 0.0138 | 0.0308 | 0.0447 | 0.0154 | 0.0320 | 0.0453 |
| ChatREC | 0.0826 | 0.1717 | 0.2308 | 0.0259 | 0.0426 | 0.0512 | 0.0382 | 0.0835 | 0.1155 | 0.0329 | 0.0729 | 0.1009 | 0.0159 | 0.0339 | 0.0477 |
| InteRecAgent | 0.0327 | 0.0719 | 0.0974 | 0.0146 | 0.0227 | 0.0264 | 0.0181 | 0.0379 | 0.0507 | 0.0125 | 0.0280 | 0.0387 | 0.0113 | 0.0205 | 0.0263 |
| $M^3$AGENT (Ours) | **0.1152** | **0.2283** | **0.2971** | **0.0448** | **0.0686** | **0.0807** | **0.0576** | **0.1185** | **0.1593** | **0.0455** | **0.0967** | **0.1308** | **0.0274** | **0.0551** | **0.0760** |
| *Methods with ideal utility model* | | | | | | | | | | | | | | | |
| Ideal utility model | 0.3502 | 0.3599 | 0.3625 | 0.1983 | 0.2041 | 0.2052 | 0.2340 | 0.2444 | 0.2472 | 0.1467 | 0.1513 | 0.1524 | 0.1588 | 0.1694 | 0.1723 |
| InteRecAgent | 0.4882 | 0.5070 | 0.5076 | 0.2171 | 0.2199 | 0.2201 | 0.2741 | 0.2951 | 0.2957 | 0.2344 | 0.2562 | 0.2568 | 0.1317 | 0.1469 | 0.1473 |
| $M^3$AGENT (Ours) | **0.7392** | **0.7629** | **0.7634** | **0.4312** | **0.4305** | **0.4305** | **0.4911** | **0.5309** | **0.5323** | **0.4311** | **0.4725** | **0.4742** | **0.2728** | **0.3093** | **0.3114** |
| **Amazon dataset** | | | | | | | | | | | | | | | |
| *Methods without ideal utility model* | | | | | | | | | | | | | | | |
| History | 0.2555 | 0.2962 | 0.3087 | 0.0621 | 0.0763 | 0.0801 | 0.1114 | 0.1423 | 0.1527 | 0.1192 | 0.1379 | 0.1429 | 0.0179 | 0.0385 | 0.0468 |
| KNN | 0.2372 | 0.3910 | 0.4709 | 0.0713 | 0.1056 | 0.1204 | 0.1127 | 0.2185 | 0.2840 | 0.1062 | 0.2108 | 0.2758 | 0.0340 | 0.0869 | 0.1283 |
| ChatREC | 0.2269 | 0.3754 | 0.4548 | 0.0630 | 0.0951 | 0.1087 | 0.1020 | 0.1985 | 0.2581 | 0.0958 | 0.1873 | 0.2452 | 0.0298 | 0.0728 | 0.1068 |
| InteRecAgent | 0.0056 | 0.0123 | 0.0178 | 0.0031 | 0.0039 | 0.0048 | 0.0030 | 0.0060 | 0.0090 | 0.0021 | 0.0046 | 0.0069 | 0.0016 | 0.0027 | 0.0041 |
| $M^3$AGENT (Ours) | **0.3267** | **0.4837** | **0.5501** | **0.1842** | **0.2113** | **0.2285** | **0.1686** | **0.2859** | **0.3417** | **0.1340** | **0.2404** | **0.2938** | **0.0736** | **0.1365** | **0.1732** |
| *Methods with ideal utility model* | | | | | | | | | | | | | | | |
| Ideal utility model | **0.6620** | 0.7292 | 0.7501 | 0.2895 | 0.3337 | 0.3450 | **0.4078** | 0.5222 | 0.5590 | **0.4082** | 0.5023 | 0.5272 | 0.1507 | 0.2987 | 0.3558 |
| InteRecAgent | 0.1852 | 0.2004 | 0.2007 | 0.0960 | 0.0880 | 0.0871 | 0.0891 | 0.1068 | 0.1082 | 0.0724 | 0.0889 | 0.0904 | 0.0351 | 0.0448 | 0.0477 |
| $M^3$AGENT (Ours) | 0.6206 | **0.7600** | **0.7733** | **0.4097** | **0.4451** | **0.4552** | 0.3569 | **0.5441** | **0.5861** | 0.3065 | **0.4891** | **0.5361** | **0.1647** | **0.3177** | **0.3770** |

The overall performance comparison results are presented in Table 1.   Our proposed $M^3$AGENT consistently and significantly outperforms all baseline methods across nearly all metrics on both the H&M and Amazon datasets, regardless of the utility model configuration.

In the more realistic scenario without the ideal utility model, $M^3$AGENT demonstrates clear superiority. On the H&M dataset, $M^3$AGENT achieves a Hit Rate@1 of 0.1152, which is a 39.5%

improvement over the strongest baseline, ChatREC (0.0826). The improvement in ranking quality is even more pronounced, with NDCG@1 showing a 73.0% increase, reaching 0.0448 compared to ChatREC's 0.0259. On the more complex Amazon dataset, our approach achieves a Hit Rate@1 of 0.3267, outperforming the best baseline (History, 0.2555) by 27.9%, and an NDCG@1 of 0.1842, exceeding the best baseline (KNN, 0.0713) by a remarkable 158.3%.

The most striking performance gains are observed in the Perfect Hit Rate metric, which measures the system's ability to satisfy all user constraints within the recommendation set. On H&M, $M^3$AGENT achieves a Perfect Hit Rate@5 of 0.0760, a 59.3% improvement over the next best method, ChatREC (0.0477). On Amazon, the metric reaches 0.1732, representing a 35.0% improvement over KNN (0.1283). These substantial gains highlight our system's superior capability in handling complex, multi-objective queries that require a complete set of items.

When integrated with an ideal utility model, the performance of $M^3$AGENT is further amplified, establishing a strong upper bound. On H&M, our method achieves a Hit Rate@1 of 0.7392, a 51.4% improvement over the next-best agent-based method, InteRecAgent (0.4882). On Amazon, $M^3$AGENT also dominates, particularly in set-based metrics like Hit Rate@5 (0.7733) and Perfect Hit Rate@5 (0.3770), far surpassing all other methods. This demonstrates our framework's exceptional ability to leverage strong utility signals to fulfill complex user needs.

Examining the baselines reveals their limitations. History-based sampling performs poorly, especially on H&M, confirming that past purchases are insufficient for specific, intent-driven queries. KNN and ChatREC show better query understanding but still fall significantly short of $M^3$AGENT, underscoring the necessity of a structured planning and optimization framework. InteRecAgent struggles notably on the Amazon dataset, suggesting its pipeline is not robust enough for diverse and complex constraints. These results validate that the architectural choices in $M^3$AGENT —combining agentic planning and multi-objective optimization—are key to its state-of-the-art performance.

## 5.3 COMPONENT-WISE ABLATION STUDY

Table 2: Ablation study: Impact of different components on recommendation performance

| | H&M Dataset | | | | | | | | | | | | | | |
| | Hit Rate@k | | | NDCG@k | | | Recall_macro@k | | | Recall_micro@k | | | Perfect Hit Rate@k | | |
| Method | @1 | @3 | @5 | @1 | @3 | @5 | @1 | @3 | @5 | @1 | @3 | @5 | @1 | @3 | @5 |
| --- | --- | --- | --- | --- | --- | --- | --- | --- | --- | --- | --- | --- | --- | --- | --- |
| *Methods without ideal utility model* | | | | | | | | | | | | | | | |
| $M^3$AGENT w/o Filtering | 0.0583 | 0.1309 | 0.1804 | 0.0235 | 0.0389 | 0.0464 | 0.0297 | 0.0671 | 0.0928 | 0.0228 | 0.0527 | 0.0741 | 0.0149 | 0.0330 | 0.0444 |
| $M^3$AGENT w/o Tree | 0.0840 | 0.1656 | 0.2166 | 0.0375 | 0.0568 | 0.0663 | 0.0455 | 0.0927 | 0.1230 | 0.0328 | 0.0684 | 0.0915 | 0.0242 | 0.0493 | 0.0656 |
| $M^3$AGENT w/o MOS | 0.0335 | 0.0775 | 0.1084 | 0.0128 | 0.0212 | 0.0262 | 0.0168 | 0.0381 | 0.0546 | 0.0128 | 0.0306 | 0.0435 | 0.0087 | 0.0178 | 0.0259 |
| $M^3$AGENT (Full) | **0.1152** | **0.2283** | **0.2971** | **0.0448** | **0.0686** | **0.0807** | **0.0576** | **0.1185** | **0.1593** | **0.0455** | **0.0967** | **0.1308** | **0.0274** | **0.0551** | **0.0760** |
| *Methods with ideal utility model* | | | | | | | | | | | | | | | |
| $M^3$AGENT w/o Filtering | 0.6063 | 0.6322 | 0.6331 | 0.3054 | 0.3032 | 0.3033 | 0.3630 | 0.3934 | 0.3947 | 0.3196 | 0.3521 | 0.3535 | 0.1775 | 0.1971 | 0.1984 |
| $M^3$AGENT w/o Tree | 0.5565 | 0.5755 | 0.5759 | 0.3333 | 0.3300 | 0.3299 | 0.3696 | 0.3981 | 0.3990 | 0.2986 | 0.3278 | 0.3288 | 0.2155 | 0.2402 | 0.2419 |
| $M^3$AGENT w/o MOS | 0.7321 | 0.7674 | 0.7691 | 0.4180 | 0.4223 | 0.4227 | 0.4791 | 0.5315 | 0.5340 | 0.4195 | 0.4735 | 0.4759 | 0.2602 | 0.3075 | 0.3103 |
| $M^3$AGENT (Full) | **0.7392** | **0.7629** | **0.7634** | **0.4312** | **0.4305** | **0.4305** | **0.4911** | **0.5309** | **0.5323** | **0.4311** | **0.4725** | **0.4742** | **0.2728** | **0.3093** | **0.3114** |
| | Amazon dataset | | | | | | | | | | | | | | |
| | Hit Rate@k | | | NDCG@k | | | Recall_macro@k | | | Recall_micro@k | | | Perfect Hit Rate@k | | |
| Method | @1 | @3 | @5 | @1 | @3 | @5 | @1 | @3 | @5 | @1 | @3 | @5 | @1 | @3 | @5 |
| *Methods without ideal utility model* | | | | | | | | | | | | | | | |
| $M^3$AGENT w/o Filtering | 0.2785 | 0.4355 | 0.5084 | 0.1567 | 0.1793 | 0.1977 | 0.1373 | 0.2450 | 0.3052 | 0.1113 | 0.2077 | 0.2640 | 0.0576 | 0.1107 | 0.1473 |
| $M^3$AGENT w/o Tree | 0.3217 | 0.4742 | 0.5395 | 0.1813 | 0.2075 | 0.2245 | 0.1650 | 0.2804 | 0.3362 | 0.1329 | 0.2366 | 0.2888 | 0.0709 | 0.1341 | 0.1714 |
| $M^3$AGENT w/o MOS | 0.0223 | 0.0422 | 0.0556 | 0.0132 | 0.0151 | 0.0172 | 0.0103 | 0.0201 | 0.0269 | 0.0089 | 0.0180 | 0.0247 | 0.0037 | 0.0069 | 0.0093 |
| $M^3$AGENT (Full) | **0.3267** | **0.4837** | **0.5501** | **0.1842** | **0.2113** | **0.2285** | **0.1686** | **0.2859** | **0.3417** | **0.1340** | **0.2404** | **0.2938** | **0.0736** | **0.1365** | **0.1732** |
| *Methods with ideal utility model* | | | | | | | | | | | | | | | |
| $M^3$AGENT w/o Filtering | 0.5703 | 0.7789 | 0.8259 | 0.3651 | 0.4249 | 0.4549 | 0.3205 | 0.5473 | 0.6372 | 0.2806 | 0.5013 | 0.5954 | 0.1410 | 0.3132 | 0.4190 |
| $M^3$AGENT w/o Tree | 0.6101 | 0.7450 | 0.7587 | 0.4044 | 0.4369 | 0.4468 | 0.3518 | 0.5320 | 0.5744 | 0.2999 | 0.4763 | 0.5228 | 0.1646 | 0.3112 | 0.3710 |
| $M^3$AGENT w/o MOS | 0.6046 | 0.7540 | 0.7744 | 0.3782 | 0.4175 | 0.4315 | 0.3269 | 0.5283 | 0.5841 | 0.2805 | 0.4750 | 0.5338 | 0.1384 | 0.2975 | 0.3733 |
| $M^3$AGENT (Full) | **0.6206** | **0.7600** | **0.7733** | **0.4097** | **0.4451** | **0.4552** | **0.3569** | **0.5441** | **0.5861** | **0.3065** | **0.4891** | **0.5361** | **0.1647** | **0.3177** | **0.3770** |

To quantify the contribution of each key module, we conducted a systematic ablation study, with results presented in Table 2. The findings clearly establish a hierarchy of importance among the components, with the Multi-Objective Search (MOS) being the most critical.

Removing the **MOS component** causes the most catastrophic performance degradation across both datasets. In the setting without an ideal utility model on the Amazon dataset, Hit Rate@1 plummets from 0.3267 to 0.0223, a staggering 93.2% decrease. The effect is even more pronounced on Perfect Hit Rate@5, which collapses by 94.6% (from 0.1732 to 0.0093). On the H&M dataset, the Hit Rate@1 drops by 70.9%. These results unequivocally demonstrate that the MOS is the cornerstone of our framework's ability to handle complex queries. Without it, the agent is unable to effectively navigate trade-offs and satisfy the set-level constraints inherent in multi-item requests, leading to a near-total failure in recommendation quality.

Removing the **candidate filtering mechanism (w/o Filtering)** also leads to a significant, albeit smaller, performance drop. On H&M, Hit Rate@1 decreases by 49.4% (from 0.1152 to 0.0583), indicating that without an effective pre-filtering of the item space, the subsequent planning and optimization steps are overwhelmed by irrelevant candidates. The impact is less severe on the Amazon dataset (a 14.8% drop in Hit Rate@1), but still substantial, confirming the value of this component in improving efficiency and focus.

The **Split-Prune Constraint Tree (w/o Tree)** shows a more nuanced impact. On the H&M dataset, its removal results in a 27.1% drop in Hit Rate@1, highlighting its role in structuring the reasoning process for fashion-related queries. However, on the Amazon dataset, the performance decrease is minimal (a 1.5% drop in Hit Rate@1). This suggests that while the Split-Prune Constraint Tree is beneficial, its importance may vary depending on the complexity and nature of the dataset and queries.

In summary, the ablation study confirms that all components contribute positively to $M^3$AGENT 's performance. The MOS is indispensable for complex constraint satisfaction, candidate filtering is crucial for efficiency, and Split-Prune Constraint Tree provides valuable structure to the agent's reasoning process.

## 5.4 FINETUNE STUDY

To explore the potential for further performance improvements, we conducted experiments by fine-tuning the underlying Large Language Model on the ground truth item attributes as constraints. The results, presented in Table 3, show that fine-tuning yields consistent, albeit modest, gains across most metrics, particularly when the Split-prune constraint tree is included in the fine-tuning data.

Table 3: Finetune results on H&M and Amazon datasets

| | H&M Dataset | | | | | | | | | | | | | | |
| --- | --- | --- | --- | --- | --- | --- | --- | --- | --- | --- | --- | --- | --- | --- | --- |
| | Hit Rate@k | | | NDCG@k | | | Recall macro@k | | | Recall micro@k | | | Perfect Hit Rate@k | | |
| Method | @1 | @3 | @5 | @1 | @3 | @5 | @1 | @3 | @5 | @1 | @3 | @5 | @1 | @3 | @5 |
| *Methods without ideal utility model* | | | | | | | | | | | | | | | |
| Finetune w/o Tree | 0.1120 | 0.2300 | 0.2990 | 0.0415 | 0.0663 | 0.0776 | 0.0573 | 0.1200 | 0.1616 | 0.0446 | 0.0975 | 0.1337 | 0.0240 | 0.0560 | 0.0770 |
| Finetune w/ Tree | **0.1160** | **0.2430** | **0.3150** | 0.0435 | **0.0695** | **0.0808** | **0.0594** | **0.1277** | **0.1686** | **0.0476** | **0.1063** | **0.1417** | **0.0280** | **0.0610** | **0.0810** |
| $M^3$AGENT (Ours) | 0.1152 | 0.2283 | 0.2971 | **0.0448** | 0.0686 | 0.0807 | 0.0576 | 0.1185 | 0.1593 | 0.0455 | 0.0967 | 0.1308 | 0.0274 | 0.0551 | 0.0760 |
| *Methods with ideal utility model* | | | | | | | | | | | | | | | |
| Finetune w/o Tree | 0.7230 | 0.7470 | 0.7480 | 0.4224 | **0.4334** | **0.4334** | 0.5140 | 0.5439 | 0.5444 | 0.4541 | 0.4827 | 0.4834 | **0.3140** | **0.3420** | **0.3420** |
| Finetune w/ Tree | **0.7750** | **0.7950** | **0.7960** | 0.4196 | 0.4306 | 0.4307 | **0.5298** | **0.5603** | **0.5611** | **0.4789** | **0.5090** | **0.5101** | 0.2910 | 0.3190 | 0.3210 |
| $M^3$AGENT (Ours) | 0.7392 | 0.7629 | 0.7634 | **0.4312** | 0.4305 | 0.4305 | 0.4911 | 0.5309 | 0.5323 | 0.4311 | 0.4725 | 0.4742 | 0.2728 | 0.3093 | 0.3114 |

**Analysis of Finetune Study** On the H&M dataset, the fine-tuned model that includes the Split-Prune Constraint Tree ( "Finetune w/ Tree ") achieves the best performance in the majority of metrics. For instance, in the scenario without the ideal utility model, it improves the Hit Rate@5 from 0.2971 to 0.3150, a 6.0% increase over the base $M^3$AGENT . Similarly, when using the ideal utility model, the Hit Rate@1 sees a notable jump from 0.7392 to 0.7750, an improvement of 4.8%. This suggests that fine-tuning helps the model better internalize the structured constraints inherent in the Split-Prune Constraint Tree. Interestingly, fine-tuning without the Split-Prune Constraint Tree ( "Finetune w/o Tree ") shows mixed results, even slightly underperforming the base model in some key metrics like Hit Rate@1. This indicates that simply fine-tuning on constraints without the underlying planning logic is less effective and can even be detrimental.

## 6 CONCLUSION

In this work, we introduced $M^3$AGENT , a novel agentic framework designed to solve the critical challenge of complex, multi-item, multi-constraint conversational recommendation. Moving beyond brittle pipelines and ungrounded LLMs, our approach represents a paradigm shift to active, goal-oriented planning. By reformulating the task as a unified multi-objective optimization problem, our agent leverages an Split-Prune Constraint Tree for grounded memory and a Pareto-complete search to deliver plans that are not only feasible but also provably optimal within the defined solution space. The significance of this work extends beyond a single application, presenting a new model for building rational, trustworthy, and deeply grounded conversational AI systems. This work marks a significant step toward a new generation of agents that are not only linguistically fluent but also demonstrably rational in complex, real-world environments.

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

## A  GLOBAL CONSTRAINT FORMULATION

In our framework, global constraints are those that apply to the recommendation tuple $y = (y_1, \ldots, y_n)$ as a whole, rather than to individual items. These are formally collected in the set $\mathcal{G}(q)$, defined as:

$$
\mathcal{G}(q) = \Big\{ y \in \mathcal{P}^n : \underbrace{\sum_{j=1}^{n} \phi_k(y_j) \le B_k}_{\text{numeric (e.g., budget)}}, \; \underbrace{L_\ell \le \sum_{j=1}^{n} \mathbf{1}_{\{y_j \in S_\ell\}} \le U_\ell}_{\text{quotas (e.g., brand diversity)}}, \; \ldots \Big\}.
$$

This structure is highly flexible and can capture a wide range of common user requirements. Common examples we handle include:

- **Budget:** The sum of the prices of all items must not exceed a total budget $B$. This is captured by setting $\phi_{\text{price}}(i) = \text{price}(i)$ and $B_{\text{price}} = B$.

- **Quotas:** The number of items from a specific set $S_\ell$ (e.g., a particular brand or category) must be within a lower bound $L_\ell$ and an upper bound $U_\ell$.

- **Delivery Deadline:** The latest delivery date among all items must not exceed a deadline $D$, i.e., $\max_j \text{ETA}(y_j) \leq D$.

- **Total Weight/Dimensions:** The sum of weights or other physical properties must not exceed a certain limit, useful for shipping constraints.

## B    PROOF OF THEOREM 4.3

**Theorem B.1** (Computational Efficiency). *Let the size of the feasible set derived from the constraint tree be bounded by $M = |\mathcal{Y}|$. The overall time complexity of our pipeline to compute the Pareto-optimal set $\mathcal{P}^\star$ is $O(N + M)$, where $N$ is the total number of items in the catalog. This represents an exponential speedup over exhaustive search methods, which have a complexity of $\Omega(N^n)$.*

*Proof.* The proof analyzes the two primary phases of the procedure: constraint-induced space reduction and the subsequent search over this reduced space.

*Constraint-Induced Space Reduction.* The initial phase constructs a candidate set $C_j(q) \subseteq \mathcal{P}$ for each of the $n$ item positions. This is achieved by scanning the full catalog $\mathcal{P}$ (of size $N$) against a constant number of attribute predicates inferred from the query. This filtering step requires $O(N)$ time per item position, leading to a total time of $O(N)$ for this phase since $n$ is a small constant. This phase only populates the candidate sets and does not enumerate their combinations.

*Search on the Reduced Space.* The second phase operates on the feasible set $\mathcal{Y}$, which is constructed from combinations of the candidate sets, i.e., $\mathcal{Y} \subseteq \prod_{j=1}^{n} C_j(q)$. The size of this search space is therefore bounded by $M = \prod_{j=1}^{n} |C_j(q)|$. Let $S(x)$ be the time complexity of the multi-objective search routine on a space of size $x$. The algorithm explores combinations only within this reduced space, so the complexity of this phase is $S(M)$. The total time complexity of our unified procedure is thus $T_{\text{uni}}(q) = O(N + S(M))$.

*Comparison and Exponential Speedup.* In contrast, any exhaustive method that explores the full unconstrained space of $\mathcal{P}^n$ must consider $N^n$ possible tuples. For any monotone search routine whose running time is non-decreasing in the number of states, its complexity is lower-bounded by the work needed to evaluate these tuples. Thus, the time complexity of an exhaustive search is $T_{\text{exh}}(N, n) = \Omega(S(N^n))$.

For the common case where the search cost is proportional to the number of tuples evaluated (i.e., $S(x) = \Theta(x)$), our procedure's complexity becomes $T_{\text{uni}}(q) = O(N + M)$, as stated in the theorem. To quantify the speedup, let the pruning fraction for each item be $\alpha_j \in (0, 1]$, such that $|C_j(q)| = \alpha_j N$. The size of the reduced space is then $M = (\prod_{j=1}^{n} \alpha_j)N^n$. When the query constraints are informative, each $\alpha_j \ll 1$, and the factor $\prod_{j=1}^{n} \alpha_j$ provides an exponential reduction in the dominant term compared to the $\Omega(N^n)$ complexity of exhaustive search. $\qquad\square$

## C    PROOF OF THEOREM 4.4

**Theorem C.1** (Pareto-Completeness of Search). *Let $\mathcal{Y}$ be the finite feasible set of tuples derived from Algorithm 1. The set $\mathcal{P}^\star$ returned by Algorithm 2 contains every Pareto-optimal tuple $y^\dagger \in \mathcal{Y}$ with respect to the objectives $(R, U, H)$.*

*Proof.* The proof proceeds by demonstrating that the algorithm systematically explores the solution space in a manner that guarantees the discovery of every Pareto-optimal point.

Fix an arbitrary Pareto-optimal tuple $y^\dagger \in \mathcal{Y}$. Let its objective vector be $(R(y^\dagger), U(y^\dagger), H(y^\dagger))$. Define its $H$-level as $\epsilon^\dagger := H(y^\dagger)$.

Consider the subset of solutions $S := \{y \in \mathcal{Y} : H(y) \geq \epsilon^\dagger\}$. By definition, $y^\dagger \in S$. The Pareto-optimality of $y^\dagger$ over the full set $\mathcal{Y}$ with respect to $(R, U, H)$ implies that there exists no other solution $y \in S$ such that $(R(y), U(y)) \succeq (R(y^\dagger), U(y^\dagger))$ with at least one strict inequality. If such a $y$ existed, it would also satisfy $H(y) \geq H(y^\dagger)$, thus dominating $y^\dagger$ in the three-objective space, which contradicts our initial assumption. Therefore, $y^\dagger$ is necessarily Pareto-optimal for the bi-objective problem over $(R, U)$ within the restricted set $S$.

Algorithm 2 iterates through every distinct $H$-value present in $\mathcal{Y}$, so it will execute a loop for $\epsilon = \epsilon^\dagger$. In this iteration, it calls the procedure COMPUTEPARETOFRONT on the set $S$. While Algorithm 2 presents this step abstractly for clarity, its implementation relies on the provably-complete $\epsilon$-constraint method to find the entire bi-objective Pareto set. For a finite set, this classical method guarantees completeness by solving two series of single-objective subproblems:

(i) $\max R(y)$ subject to $y \in S$ and $U(y) \geq \tau$, for every distinct value $\tau$ attained by solutions in $S$.

(ii) $\max U(y)$ subject to $y \in S$ and $R(y) \geq \rho$, for every distinct value $\rho$ attained by solutions in $S$.

Since $y^\dagger$ is Pareto-optimal for $(R, U)$ in $S$, it is guaranteed to be found as an optimal solution to at least one of these subproblems (specifically, when $\tau = U(y^\dagger)$ or $\rho = R(y^\dagger)$).

Thus, the call to COMPUTEPARETOFRONT will identify $y^\dagger$ and include it in the set $\mathcal{P}_\epsilon$. The subsequent update procedure, UPDATEGLOBALPARETOSET, will correctly insert $y^\dagger$ into the global set $\mathcal{P}^\star$ (as it is non-dominated by any other point). Since our choice of $y^\dagger$ was arbitrary, this holds for all Pareto-optimal tuples, proving the algorithm's completeness. □

# D PROMPTS

## D.1 PROMPT: QUERY GENERATION

```
1  system = f"""You are an online shopper.
2  Tone: friendly, conversational.
3  """
4
5  user_prompt = f"""<SCENE>
6  You just added these items to your cart:
7
8  {chr(10).join(items)}
9
10 Total budget: ${budget:.0f}
11 ---
12 TASK:
13 Repeat (i.e. restate) the natural-language query you originally typed.
14 MUST: mention your exact overall budget naturally. mention the exact
       maximum acceptable price for a single item if and only if it is
       provided; refer to product attributes (type, color, appearance,
       style, material etc.) without copying exact product details.
15
16 Please respond with **only** your shopper query.
17 """
```

## D.2 PROMPT: EXTRACT CONSTRAINTS

```
1  system: |
2    You are a constraint extraction component of an intelligent
       shopping assistant. Analyze the user input, determine if it is
       expressing a shopping need. If no, directly return None.
```

```
3       If yes, extract the constraints or update the current constraints
        if provided.
4
5       Complete constraints include:
6       1. Total budget (if specified)
7       2. Items needed represented as categorical constraints and price (
        if specified)
8
9       IMPORTANT: For categorical constraint, select ONLY valid field and
         values from the list:
10      {catalog_attributes}
11
12      Output in JSON format:
13      {{
14        "budget": budget(number),
15        "items": [
16          {{
17            "constraint1": ["value"] ,
18            "constraint2": ["value1", "value2"],
19            "price_max": price(number)
20          }},
21          ... more items
22        ]
23      }}
24      <Please respond with **only** JSON>
25    user: |
26      Current constraints:
27      {existing_constraints}
28      User's recent conversation history (for context):  {chat_hist}
```

### D.3  PROMPT: ITEM DESCRIPTION GENERATION

```
1    system: |
2      You are a shopping assistant helping to clarify vague user intent.
3      Your task is to generate concise natural language descriptions for
       multiple items based on the provided constraints.
4
5      Rules:
6      - Output ONLY a valid JSON array of strings
7      - Each string corresponds to exactly one item in the input (1-to-1
       mapping, no extra or missing entries)
8      - Each description must be a single concise sentence
9      - Include inferred preferences or missing details if appropriate
10     - Do NOT include reasoning, explanations, suggestions, or
       alternatives
11     - Do NOT add any text outside the JSON array
12     - Example output: ["a short description", "another description"]
13   user: |
14     {user_info}
15     User's recent conversation history (for context): {chat_hist}
16
17     Items to describe (each item will correspond to one output string)
       :
18     {all_items_constraints}
```

## E  EVALUATION METRICS DETAILS

We evaluate our models using multiple metrics at different cutoff points (1, 3, and 5), all implemented in our evaluation_utils.py framework:

- **Hit Rate@k**: The proportion of test cases where at least one relevant item appears within the top-k recommendations.

$$HitRate_u = \begin{cases} 1, & \text{if } |Hits_u| > 0 \\ 0, & \text{otherwise} \end{cases} \tag{2}$$

where $Hits_u$ is the intersection of recommended items and ground truth items for user $u$.

- **NDCG@k**: Normalized Discounted Cumulative Gain, which measures both relevance and ranking quality.

$$NDCG_u = \frac{DCG_u}{IDCG_u}, \quad DCG_u = \sum_{j=1}^{L} \frac{2^{h_{u,j}} - 1}{\log_2(j+2)} \tag{3}$$

where $h_{u,j}$ is the number of new relevant items found in the $j$-th recommendation set, and $IDCG_u$ is the ideal DCG value when all relevant items are ranked at the top.

- **Recall_macro@k**: The average per-user recall rate, treating each user equally.

$$Recall\_macro = \frac{1}{|U|} \sum_{u \in U} Recall_u, \quad Recall_u = \frac{|Hits_u|}{|G_u|} \tag{4}$$

where $G_u$ is the set of ground truth items for user $u$.

- **Recall_micro@k**: The global recall calculated as the total number of correctly recommended items divided by the total number of relevant items.

$$Recall\_micro = \frac{\sum_{u \in U} |Hits_u|}{\sum_{u \in U} |G_u|} \tag{5}$$

- **Perfect Hit Rate@k**: The proportion of test cases where all relevant items appear within the top-k recommendations.

$$PerfectHit_u = \begin{cases} 1, & \text{if } |Hits_u| = |G_u| \\ 0, & \text{otherwise} \end{cases} \tag{6}$$

These metrics capture different aspects of recommendation performance. User needs in our evaluation framework encompass both immediate needs (expressed through queries) and profile-based preferences (derived from user history).

