# OpenReview forum: "Satisfying Complex User Needs: M^3 Agent for Conversational Multi-Item Recommendation"
_ICLR.cc/2026/Conference — Submitted to ICLR 2026_

### Official Review · Reviewer_nVkM · 2025-10-30

**Soundness:** 3
**Presentation:** 2
**Contribution:** 3
**Rating:** 4
**Confidence:** 2

**Summary:**

Existing paradigms, from traditional recommenders to modern LLM-based agents, fail due to a crisis of grounding and an inability to handle trade-off complexity. Therefore, this paper proposes an agentic framework (called M3AGENT) that bridges natural language and grounded, optimal recommendations. Experiments conducted on real-world datasets demonstrate the effectiveness of the proposed method.

**Strengths:**

1. The first work is to reformulate multi-item recommendation with constraints as a multi-objective optimization problem.

2. A novel data structure that serves as its working memory to translate unstructured language into a feasible solution space grounded in the live catalog.

3. Theoretical analysis and experiments are provided to show the effectiveness of the proposed method.

**Weaknesses:**

1. Statisitcal analysis is missing.
2.  Lack of the time complexity of the proposed method;

**Questions:**

(1)	Experiments showed the effectiveness of the proposed method. Is there a significant difference among all competing ones?
(2)	How is the time complexity of the proposed method?

---

### Official Review · Reviewer_XaL4 · 2025-10-30

**Soundness:** 1
**Presentation:** 3
**Contribution:** 2
**Rating:** 2
**Confidence:** 4

**Summary:**

This paper primarily addresses the multi-constraint problem of multi-item recommendation, proposing an agent framework to meet the complex user needs in conversational recommendation. The paper formally derives the problem, which seems interesting. However, the experimental setup is incomplete, and the chosen baselines are relatively simple. Related work is also lacking. Overall, while the problem it addresses is good and the proposed method has some innovation, it hasn't undergone sufficient comparison, and the experimental results don't demonstrate superior performance.

**Strengths:**

1. This paper proposes an agent framework to meet the complex needs of users in conversational recommendation.
2. This paper provides formal theoretical guarantees of efficiency and optimality (Pareto-completeness).

**Weaknesses:**

1. The experimental setup is incomplete, and the choice of the baseline for the recommender system is problematic. The best baseline for recommender systems is KNN. KNN is clearly not the best method.

2. Section 2.2 of the related work does not include classic papers on conversational recommendation; most of the content is related to LLM.

3. Table 1 lacks significance testing.

4. It seems somewhat exaggerated. Especially since the authors claim to be the first to formally define the recommendation of multiple constrained items as a multi-objective optimization problem. Multi-objective optimization does not seem to be a very new method.

**Questions:**

- Does "complex requirements" refer to numerous preference attributes? I feel this might not be so complex; if the preference attributes are clearly defined, it's relatively easy to implement.

- Regarding the datasets:
1. Why were these two datasets chosen?
2. "For each user, we randomly select 1-5 items from their history and use Claude-Sonnet-4 to generate user queries." Why do this? Is this reasonable? Is there any supporting evidence?


- Regarding Table 1:
1. In practice, what is the Ideal utility model? In "Methods with Ideal utility model," what does the "Ideal utility model" method mean? On which method is the Ideal utility model added?

2. Why does InteRecAgent perform particularly poorly on the Amazon dataset? Is there any supporting evidence for the explanation, "its pipeline is not robust enough for diverse and complex constraints"?

3. What is the difference between Multi-item Recommendation and next basket recommendation addressed in this article? Could we add some classic methods like next basket recommendation and sequential recommendation as baselines?

4. On the Amazon dataset, M$^3$AGENT with the ideal utility model doesn't perform as well as the ideal utility model in metrics like Hit Rate@1 Recall. Why?


- Regarding Table 2:

1. On the Amazon dataset, for Methods with the ideal utility model, the highest Hit Rate@5 is 0.8259. Why isn't this the highest result bolded? Several other results show the same issue.

2. The ablation results on the Amazon dataset seem to indicate that these modules aren't critical, while the ideal utility model is the most important part.

- Regarding Table 3:

1. The effect of fine-tuning seems insignificant; the results (for most metrics) of several methods don't show much difference. This is a bit strange; normally, fine-tuning should significantly improve performance.


- The first two pages contain many citations of "et al."; these should be enclosed in parentheses (et al.).

---

### Official Review · Reviewer_p4gK · 2025-10-31

**Soundness:** 2
**Presentation:** 2
**Contribution:** 2
**Rating:** 2
**Confidence:** 4

**Summary:**

The paper proposes M^3AGENT, a framework for conversational multi-item recommendation under multiple constraints. The core idea is to cast planning as a unified multi-objective optimization over relevance, utility, and constraint satisfaction, combining a Split-Prune Constraint Tree to define a feasible set and a Pareto-complete search to select non-dominated solutions; the authors claim theoretical guarantees for completeness and optimality within the feasible set. Experiments on two e-commerce datasets (H&M and Amazon) use LLM-generated user prompts and standard top-k metrics, with ablations suggesting the multi-objective search is critical to performance and some gains over baselines like ChatREC and InteRecAgent are reported.

**Strengths:**

- The problem in study is important and practical.

- A reasonably coherent formulation. The unified multi-objective framing and the Split-Prune Constraint Tree are a tidy way to connect free-text constraints to a feasible space, followed by Pareto-complete search.

- Some theoretical framing and ablation analysis.

**Weaknesses:**

- Misalignment between motivation and evaluation: no multi-turn conversational evaluation. Although the method is pitched as conversational and  multi-turn adaptable, the experiments appear to evaluate single-shot prompts generated from user histories, without a true multi-turn protocol, user simulation, or dialogue progression metrics. This leaves the central conversational claim under-evaluated .

- Constraint satisfaction and coverage are not measured end-to-end. While the paper defines global constraints and constructs a feasible set, it does not report any metric that quantifies how often returned plans actually satisfy all item-level and cross-item constraints extracted from language, nor the coverage of constraint-satisfying items. The reported metrics do not assess constraint adherence, so it is unclear whether constraints are consistently met in practice, especially under imperfect extraction or pruning  .

- No inference cost or latency analysis. The paper argues for efficiency conceptually, but provides no wall-clock latency, throughput, or cost breakdown for constraint extraction, tree construction, candidate filtering, and Pareto search—nor scalability with catalog size and number of requested items. For a deployment-minded setting like conversational commerce, this omission is significant.

- Synthetic, single-shot prompts limit ecological validity. The datasets use LLM-generated queries from historical items, not real user dialogues; prompts and constraint extraction templates are hand-crafted in the appendix. Without real conversations, user studies, or a multi-turn simulator, it is hard to assess robustness to messy inputs and evolving goals common in real sessions.


- Baseline and claims gaps. The baseline set does not convincingly test multi-turn constraint handling, and it is unclear whether stronger conversational agents (configured for multi-turn negotiation and constraint repair) were considered. Additionally, the paper repeatedly positions itself as “first” to unify multi-item recommendation as multi-objective optimization, yet related work has explored multi-objective and Pareto fronts in recommendation domain.

**Questions:**

Please see the details in my weaknesses section, particularly on conversational evaluation, constraint adherence, and scalability.

---

### Official Review · Reviewer_X8Aj · 2025-11-01

**Soundness:** 3
**Presentation:** 3
**Contribution:** 3
**Rating:** 4
**Confidence:** 4

**Summary:**

The paper addresses the compelling challenge of multi-item, multi-constraint conversational recommendation, correctly identifying the failures of existing LLM agents related to grounding and principled trade-off negotiation. The proposed M3AGENT framework introduces the novel concept of reformulating this task as a unified Multi-Objective Optimization (MOO) problem. However, the comprehensive ablation studies reveal that the system's success is critically dependent on its most complex component (MOS) and that the core grounding mechanism (SPCT) exhibits inconsistent efficacy across domains. Furthermore, the reliance on high-quality, external modeling for practical utility, combined with the extreme brittleness observed when core components fail, suggests that M3AGENT is not sufficiently robust for real-world deployment.

**Strengths:**

1. Novel Unified Optimization Formulation: M3AGENT provides a necessary conceptual advancement by being the first to reformulate the complex conversational multi-item recommendation task as a unified multi-objective optimization problem. This addresses the previously identified "crisis of trade-off and control" by explicitly optimizing for Relevance (R), Query-agnostic Utility (U), and Constraint Satisfaction (H) simultaneously.

2. Strong Theoretical Foundations for Efficiency: The paper provides formal theoretical guarantees of efficiency, proving that the system's overall time complexity is O(N+M) (where N is catalog size and M is the size of the feasible set). This demonstrates an exponential speedup over traditional exhaustive search methods, which have a complexity of Ω(N^n), provided that the Split-Prune Constraint Tree effectively prunes the search space.

3. Superior Constraint Satisfaction in Controlled Settings: Despite the fragility, the framework, when fully operational, exhibits superior capability in fulfilling complex, set-level constraints. This is validated by the strong results in the Perfect Hit Rate metric, where M3AGENT significantly outperforms all baselines on both the H&M (59.3% improvement over the next best) and Amazon datasets

**Weaknesses:**

1. Extreme Brittleness and Over-reliance on Multi-Objective Search (MOS): The ablation study highlights a catastrophic structural fragility in the framework. Removing the Multi-Objective Search (MOS) component causes a near-total collapse in performance, indicating that the system lacks any robust fallback mechanism for constraint satisfaction. For instance, on the Amazon dataset (without the ideal utility model), Hit Rate@1 plummets by a staggering 93.2% (from 0.3267 to 0.0223) when MOS is removed. Even more critically for a system designed to satisfy sets of constraints, the Perfect Hit Rate@5 collapses by 94.6% (from 0.1732 to 0.0093). This extreme sensitivity demonstrates that the system is not robustly agentic; rather, its efficacy is completely tied to a single, complex optimization step, failing the crucial test of reliability.

2. Inconsistent Efficacy of the Split-Prune Constraint Tree (SPCT): The Split-Prune Constraint Tree (SPCT) is presented as the novel mechanism to overcome the "grounding and brittleness crisis" by ensuring plans are "rooted in reality". However, the ablation study shows its necessity is highly dataset-dependent. While removing the SPCT ("w/o Tree") results in a significant 27.1% drop in Hit Rate@1 on the H&M dataset, the performance decrease is minimal—only 1.5% in Hit Rate@1—on the Amazon dataset. This stark inconsistency undermines the claim that the SPCT is a universally required architectural component for robust multi-item recommendation, raising questions about its generalizability across different e-commerce domains and catalog complexities.

3. Untested Vulnerability to Upstream LLM Failures: The entire planning pipeline is predicated on the initial accuracy of the Large Language Model (LLM) used to extract constraints C and generate item descriptions for Relevance R. While the SPCT attempts to mitigate LLM hallucinations via confidence scores $w_c$ and pruning, the process of relaxation relies on removing the constraint with the "lowest confidence score". The quality of the final solution thus remains vulnerable to the LLM's initial parsing errors, especially the potentially arbitrary confidence assigned to complex or ambiguous natural language inputs. The paper acknowledges that translating natural language into structured constraints is non-trivial but does not robustly evaluate the system's performance degradation when the upstream LLM is intentionally provided with ambiguous or faulty prompts.

4. Practical Performance Overwhelmingly Dependent on External Utility Model: The experiments establish a massive gap between performance without and with the "Ideal utility model". For instance, the Hit Rate@1 on H&M jumps from 0.1152 (without ideal utility) to 0.7392 (with ideal utility). This six-fold increase demonstrates that the framework's high-end performance is overwhelmingly driven by the exogenous quality of the Utility (U) model. Given that M3AGENT's own architecture does not contribute this utility modeling, and its performance without an ideal utility model is comparatively modest (e.g., Hit Rate@1 of 0.1152 on H&M, compared to the upper bound of 0.7392), the practical contribution of the M3AGENT framework itself is heavily diluted by this dependency.

**Questions:**

NA

---

### Meta-Review · Area_Chair_hw8g · 2026-01-08

**Summary:**

The paper introduces M^3Agent, a framework designed to handle complex, multi-item requests in conversational commerce through a Split-Prune Constraint Tree and Pareto-complete search. The authors aim to formulate recommendation as a unified multi-objective optimization problem to bridge natural language with grounded recommendations.

While the problem space is relevant, the reviewers identified critical flaws that undermine the paper's contributions, including the disconnect between Claims and Evaluation, missing critical metrics, restricted baselines. Due to these weaknesses, the submission is not ready for publication at ICLR.

**Reviewer Concerns:**

Most concerns cannot be addressed in a rebuttal.

**Reviewer Scores:**

Unlike

---

### Decision · Program_Chairs · 2026-01-26

Reject